# Drug–food and drug–alcohol interactions: Pharmacists' knowledge gaps and patient safety concerns in the United Arab Emirates

**Taima Qudah**[1,2], **Leen Fino**[3], **Razan I. Nassar**[3], **Suhad Abumueis**[4]*

1 Department of Clinical Pharmacy, College of Pharmacy, Al-Ain University, Abu Dhabi, United Arab Emirates, 2 Al Ain University Health and Biomedical Research Center, Al-Ain University, Abu Dhabi, United Arab Emirates, 3 Department of Clinical Pharmacy and Therapeutics, Faculty of Pharmacy, Applied Science Private University, Amman, Jordan, 4 Department of Clinical Nutrition and Dietetics, Faculty of Applied Medical Sciences, The Hashemite University, Zarqa, Jordan

* suhad.abumweis@hu.edu.jo

## Abstract

Pharmacists play a critical role in promoting medication safety through patient counselling and identification of potential interactions. However, knowledge regarding drug–food interactions (DFIs) and drug–alcohol interactions (DAIs) among community pharmacists in the United Arab Emirates (UAE) has not been previously examined. This cross-sectional study assessed pharmacists' knowledge of DFIs, DAIs, and drug–food administration timing, and identified factors associated with higher knowledge levels using a validated self-administered 24-items questionnaires. The survey assessed sociodemographic and professional characteristics, knowledge of DFIs, DAIs, and drug-food timing. Descriptive statistics and multiple linear regression analyses were performed using SPSS v28.0 to determine predictors of total knowledge scores. Pharmacists demonstrated moderate knowledge, with mean scores of 8.62/12 for DFIs, 5.84/12 for drug–food timing, and 4.70/6 for DAIs (total score 19.16/30). High awareness was observed for interactions such as caffeine–diazepam (76.3%) and methotrexate–alcohol (81.0%), whereas only 25.9% identified potassium-rich foods' interaction with spironolactone. Higher knowledge scores were associated with being male (p < 0.001), aged 20–29 years (p = 0.041), having ≥6 years of experience (p = 0.004), and higher self-perceived knowledge (p < 0.001). These findings highlight important knowledge gaps that may affect patient safety. Targeted continuing education and integration of DFI/DAI content into pharmacy curricula are recommended to strengthen counselling practices and improve medication safety outcomes.

**Data availability statement:** All relevant data are within the paper and its Supporting Information files.

**Funding:** The author(s) received no specific funding for this work.

**Competing interests:** The authors have declared that no competing interests exist.

## Introduction

The interaction between food, alcohol, and medications is a crucial yet often over-looked aspect of patient safety. Food has traditionally been valued for its nutritional content as well as its ability to prevent and treat diseases [1]. However, it has the potential to alter drug pharmacokinetics and pharmacodynamics, leading to drug–food interactions (DFIs) [2]. The U.S. Food and Drug Administration (FDA) defines DFIs as situations in which food or beverages modify the absorption, metabolism, distribution, or excretion of drugs, thereby influencing therapeutic efficacy and safety [3–5].

Similarly, drug-alcohol interactions (DAIs) pose a substantial concern, since their combination can lead to considerable adverse effects or a marked reduction in the efficacy of the treatments involved [6]. Although alcohol consumption in the United Arab Emirates is regulated, evidence indicates that alcohol use remains present and represents a public health concern in the country [7], underscoring the importance of awareness of potential drug–alcohol interactions in clinical practice. For example, Interactions between alcohol and antihistamines or nonsteroidal anti-inflammatory medicines (NSAIDs) may increase the risk of internal bleeding and respiratory complications. Alcohol may also cause life-threatening cardiovascular complications when combined with monoamine oxidase inhibitors (MAOIs) [6]. Likewise, drinking alcohol while taking psychiatric or antihypertensive drugs may result in hepatic toxicity, heart rate fluctuations, fainting, and drowsiness [6].

Drug-food Interactions and DAIs are common and clinically significant, contributing to hospitalizations, toxicity, and increased healthcare costs, particularly among older adults [8,9]. Grapefruit juice, for instance, significantly increases medication exposure and the risk of adverse effects by inhibiting CYP3A enzymes [3,10,11]. While excessive intake of vitamin K–rich foods diminishes warfarin's safety and effectiveness [6,12]. Another well-documented example is the interaction between tyramine-rich meals and MAOIs, which can cause fatal hypertensive crises [6,12]. Despite the existence of well-characterized examples, awareness among healthcare professionals remains suboptimal.

Pharmacists, as accessible healthcare providers, are uniquely positioned to identify and prevent harmful interactions through patient counselling and medication review [13]. According to international guidelines, healthcare providers should proactively detect and avoid potentially harmful interactions and give patients individualized dietary guidance [3,14]. Nevertheless, global evidence indicates that pharmacists' knowledge of DFIs and DAIs varies widely and is frequently inadequate [15–18].

To date, there haven't been many studies that evaluate pharmacists' understanding of DFIs and DAIs, and none that involve community pharmacists in the United Arab Emirates (UAE). Given their central role in patient education and medication safety, assessing their knowledge is essential. Guided by the premise that pharmacists' demographic and professional characteristics may influence their knowledge and counselling practices, we hypothesized that knowledge levels would vary according to age, gender, and years of experience. Identifying these variations may help inform targeted educational strategies and professional development initiatives to enhance medication safety. Therefore, this study aimed to evaluate the knowledge

of community pharmacists in UAE regarding DFIs and DAIs, and the timing of drug administration with respect to food, as well as to determine the variables associated with their levels of knowledge.

## Method

### Ethical considerations

In this study, human participants were prospectively recruited. The recruitment period was conducted between **March 1 and July 31, 2025**. Eligible participants were registered pharmacists holding at least a bachelor's degree in pharmacy and currently employed in community pharmacy practice in the United Arab Emirates. Pharmacy interns, students, and hospital pharmacists were excluded.

All participants **provided informed consent** prior to participation. Written informed consent was obtained electronically through the survey platform before respondents could proceed to the questionnaire. Participation was voluntary and anonymous, with no collection of personal identifiers or geolocation data. For this study, as all participants were adults, consent from parents or guardians was not applicable. The study protocol, including the consent process, was reviewed and approved by the Ethics Committee of Al Ain University (Ref. No.: COP/AREC/AD/14).

### Study design and setting

A validated, cross-sectional online survey was conducted to assess community pharmacists' knowledge of drug–food interactions (DFIs), drug–alcohol interactions (DAIs), and the timing of drug administration with respect to food in the United Arab Emirates (UAE). The study targeted licensed pharmacists employed in community pharmacies across all seven emirates.

### Study population and sampling

Eligible participants were registered pharmacists holding at least a bachelor's degree in pharmacy (or higher qualification) who were currently employed in community pharmacy practice. Pharmacy interns, students, and hospital pharmacists were excluded.

To collect data, the questionnaire was uploaded to Google Forms, an online survey platform. The inclusion criteria for the respondents were holding a bachelor's degree in pharmacy or having a higher educational qualification. An invitation to participate in the study reached the target population online, using the social media platforms of pharmacists' professional groups. This convenience-based online recruitment approach may have preferentially attracted pharmacists who are more professionally engaged or academically inclined, potentially introducing selection bias. The invitation included a brief description of the study and a link to the survey. Participants were informed that the survey was anonymous, no personal identifiers or geolocation data were collected. All responses were kept confidential. The completion of the questionnaire was voluntary, and participants could withdraw from the study at any time. A minimum sample size of 381 participants was recommended, according to the RaoSoft® calculator, based on a margin of error of 5%, confidence level of 95%, population size: 11,153 registered pharmacists In UAE [14] and a response distribution of 50%. The number of respondents included in this study was 401 registered community pharmacists.

### Instrument development and validation

The study instrument was developed following an extensive review of the literature on pharmacists' knowledge of DFIs and DAIs [17,19,16]. Questions were selected based on their comparability with prior validated tools and relevance to UAE pharmacy practice. The preliminary version was reviewed by three academic pharmacists and two community pharmacy practitioners to ensure clarity, content validity, and applicability.

The questionnaire was pilot tested among 20 pharmacists; feedback was used to refine question wording and structure. Data from the pilot phase were excluded from the final analysis. Internal consistency reliability was confirmed using Cronbach's alpha ($\alpha = 0.77$), indicating acceptable reliability.

## Study instrument

The final questionnaire contained 45 items divided into four sections:

1. **Sociodemographic and professional characteristics** (15 items): including gender, location, education level, type of pharmacy, years of experience, number of staff, presence of telepharmacy, and participation in DFI-related events.

2. **Knowledge of DFIs** (12 items): assessing awareness of well-known food–drug interactions using "Yes/No" response options.

3. **Knowledge of drug–food timing** (12 items): respondents indicated appropriate timing (e.g., 30 minutes before meal, with meal, two hours after meal, or regardless of food).

4. **Knowledge of DAIs** (6 items): pharmacists indicated whether alcohol interacts with listed medications ("Yes" for known interaction, "No" for none).

Additionally, pharmacists were asked about their main sources of DFI information, and their perceptions of which patient groups are most susceptible to DFIs.

The questionnaire primarily assessed factual knowledge using closed-ended items to ensure standardization and objective scoring across respondents. While this approach allows efficient evaluation of baseline knowledge, it may not fully reflect real-world counselling behaviors or clinical decision-making.

The full questionnaire and anonymized raw dataset are provided as Supporting Information files (S1 Questionnaire and S1 Data).

## Data collection

The final survey was uploaded on Google Forms to facilitate nationwide participation. The invitation link, including a brief description of the study, was distributed through pharmacists' professional groups and online platforms. Responses were stored securely and automatically recorded, ensuring confidentiality.

All survey questions were mandatory to prevent missing data. Incomplete questionnaires were excluded from analysis.

## Scoring and outcome measures

Each correct knowledge response was scored as "1," and incorrect or "unsure" responses were scored as "0." Total knowledge scores ranged from 0 to 30, with higher scores indicating better knowledge. Subscale scores were also calculated for each domain: DFIs (0–12), DAIs (0–6), and drug–food timing (0–12).

All items were weighted equally to maintain consistency and comparability with previous knowledge-based surveys. However, we acknowledge that different interactions may carry varying levels of clinical risk, and future research could consider risk-based weighting or case-based assessment approaches.

## Statistical analysis

After data collection, the survey responses were coded and input into a customized database using the Statistical Package for the Social Sciences (SPSS), Version 28.0 (IBM Corp., Armonk, New York, USA). Descriptive results were presented using the mean and standard deviation for the continuous variables, while the qualitative variables were presented as frequencies and percentages.

Independent-samples t-tests were conducted to compare mean knowledge scores across binary demographic and professional variables, including gender, age group, geographical location, pharmacy type, and years of experience.

Variables with $p < 0.25$ in univariate analysis were subsequently entered into a multiple linear regression model to identify independent predictors of total knowledge scores. Multicollinearity was assessed prior to model inclusion. Statistical significance was set at $p < 0.05$.

## Results

### Demographic characteristics

A total of 401 pharmacists were recruited in the current study, with more than half being females (56.9%). Most participants were aged 20–39 years (44.4% aged 20–29; 45.1% aged 30–39). Most of the participants resided in Abu Dhabi (63.1%), followed by Dubai (23.2%). A high percentage of the participants (97.3%) held a Bachelor of Pharmacy degree (Table 1).

In terms of practice setting, about two-thirds of the study's participants were practicing in chain pharmacies, while 34.2% reported practicing in individual pharmacies. Years of practice showed variations; more than half of the pharmacists (51.6%) had 3–5 years, 20.4% had 6–10 years, and 20.0% had ≥ 2 years. Pharmacists were asked to self-assess their personal information about DFIs; 70.8% reported sufficient knowledge, while 29.2% indicated a lack of knowledge in that area (Table 1).

Regarding the pharmacy work side, the majority of the participants reported that their pharmacy offers teleservices (96.5%). In addition, 87.0% of the participants reported that the number of licensed staff per shift in the pharmacy is three or fewer, while 13.0% reported having more than three pharmacists in each shift.

Study participants were asked about the age group that is most susceptible to food-drug interactions (Fig 1). The Elderly were reported the most (78.8%), while children were reported the least (23.4%).

Study participants were asked about their main source of knowledge about food-drug interactions (Fig 2). University education was reported the most (89.8%), while published literature was reported the least (23.7%).

About 39.0% of the study participants reported that it is mandatory to attend conferences, workshops, seminars, or courses related to drug-food interaction. Regarding the number of such events they had attended, 31.4% reported attending 1–2 events, 22.7% reported attending 3–5 events, and 7.0% reported attending more than five events. On the other hand, 38.9% never participated in any related events.

### Knowledge of drugs-food interactions

Fig 3 shows the percentages of correct compared to incorrect answers to the 12 knowledge items on drug-food interactions. Eleven of the items were answered correctly by more than half of the participants. The highest percentage of correct answers (76.3%) was on "*Does Caffeine consumption affect the efficacy of Diazepam?*".This was followed by "*Do protein-rich foods affect the efficacy of levodopa?*", and "*Does Cauliflower consumption affect the efficacy of Levothyroxine?*" (74.6%, and 74.3%, respectively).

The following four more items were answered correctly by 74.1%− 73.6% of the participants; "*Does the wheat bran diet affect the efficacy of Digoxin*?", "*Does milk affect the efficacy of Tetracycline"? "Patients taking Theophylline should avoid excessive coffee and tea*", and "*Patients taking Monoamine Oxidase Inhibitors (MAOIs) should avoid eating aged cheeses*". In addition, taking Atorvastatin with grapefruit was answered correctly by 69.3%, whereas taking Amiodarone with grapefruit was answered correctly by 68.8%.

On the other hand, the only question that was answered incorrectly by most of the pharmacists was "*Patients should avoid taking Spironolactone with food rich in Potassium*", as only 25.9% knew that those patients should avoid potassium-enriched foods.

**Table 1. Demographic characteristics of pharmacists (n = 401).**

| Parameter | n (%) |
|---|---|
| **Gender** | |
| Male | 173 (43.1) |
| Female | 228 (56.9) |
| **Age** | |
| 20-29 years | 178 (44.4) |
| 30-39 years | 181 (45.1) |
| 40-49 years | 28 (7.0) |
| 50-59 years | 10 (2.5) |
| >60 years | 4 (1.0) |
| **Emirate** | |
| Abu Dhabi | 253 (63.1) |
| Ajman | 14 (3.5) |
| Dubai | 93 (23.2) |
| Ras Al Khaimah | 2 (0.5) |
| Sharjah | 37 (9.2) |
| Umm Al-Quwain | 2 (0.5) |
| **Nationality** | |
| Emirati (United Arab Emirates) | 1 (0.2) |
| Syrian | 43 (10.7) |
| Jordanian | 35 (8.7) |
| Palestinian | 16 (4.0) |
| Egyptian | 134 (33.4) |
| Sudanese | 22 (5.5) |
| Indian | 95 (23.7) |
| Filipinos | 18 (4.5) |
| Pakistani | 17 (4.2) |
| British (UK) | 1 (0.2) |
| Iranian | 4 (1.0) |
| Iraqi | 10 (2.5) |
| Bangladeshis | 4 (1.0) |
| Other | 1 (0.2) |
| **Educational level** | |
| Bachelor of Pharmacy | 390 (97.3) |
| Doctor of Pharmacy (Pharm D) | 5 (1.2) |
| Postgraduate studies | 6 (1.5) |
| **Where do you practice?** | |
| Chain Pharmacy | 264 (65.8) |
| Individual Pharmacy | 137 (34.2) |
| **How many years have you been practicing?** | |
| ≥ 2 years | 80 (20.0) |
| 3-5 years | 207 (51.6) |
| 6-10 years | 82 (20.4) |
| 11-15 years | 25 (6.2) |
| >15 years | 7 (1.7) |

*(Continued)*

**Table 1.** (Continued)

| Parameter | n (%) |
|---|---|
| **Do you think you have enough information about food-drug interactions?** | |
| No | 117 (29.2) |
| Yes | 284 (70.8) |

Analysis of the drug-food interaction items revealed that the pharmacists had a mean score of 8.62 (SD = 3.09), with a minimum score of 2 out of 12, and a maximum score of 12. Notably, more than one-third of the pharmacists (38.7%) answered all the items correctly, achieving a score of 12.

### Knowledge of drug-food timing

Fig 4 shows the participants responses to the 12 items assessing knowledge about the timing of drug intake with respect to food. The highest percentage of correct answers was for "*Best time to take Omeprazole with respect to food*", with 67.6% of the pharmacists correctly identifying that it should be taken within 30 minutes before a meal. This was followed by "*Best time to take Glipizide with respect to food*", with 62.1% of the pharmacists correctly identifying that it should be taken within 30 minutes before a meal. Similarly, 61.8% of the participants correctly recognized that Levothyroxine should be taken on an empty stomach, preferably 30 minutes before a meal.

For other medications (Fig 4), participants correctly identified that Metformin, NSAIDs, Isotretinoin, Calcium carbonate supplements, and Griseofulvin should be taken with a meal (58.4%, 55.6%, 54.9%, 54.1%, and 53.1%, respectively).

Conversely, low percentages of correct answers were reported for Erythromycin stearate and Methotrexate (Fig 4).

Analysis of the knowledge about the timing of drug intake with respect to food revealed a mean score of 5.84 (SD = 3.52), with scores ranging from 0 to 11 out of 12. Although no participant correctly identified the timing for all medications; more than one-third of the pharmacists (37.4%) answered 11 out of 12 items correctly.

### Knowledge of drugs-alcohol interactions

Fig 5 illustrates the pharmacists' responses to the 6 items addressing drug-alcohol interactions. Regarding the correct answers, the highest recognition percentage was for Methotrexate and alcohol interaction, with 81.0% of pharmacists identifying it correctly. This was closely followed by Warfarin and alcohol interaction, which was acknowledged by 79.6% of participants.

Additionally, 78.3% of the participants were aware of the interaction between Isoniazid and alcohol. This percentage was also the same for Antihistamine and alcohol (Fig 5). Furthermore, more than three-quarters of the participants recognized the interaction between alcohol, and Metformin (76.8%), as well as alcohol and Paracetamol (76.1%).

Analysis of the knowledge about addressing drug-alcohol interactions revealed a mean score of 4.70 (SD = 2.13), with scores ranging from 0 to 6 out of 6. Notably, more than two-thirds of the pharmacists (68.3%) answered correctly all items about alcohol interaction.

### Participant's total knowledge scores

Exploring the results from another perspective, pharmacists had a mean total knowledge score of 19.16 (SD = 7.39) out of a possible 30, with individual scores ranging from 7 to 28. Although no participant achieved a perfect score of 30, 34.9% of pharmacists scored 28, indicating that a substantial proportion had near-maximal knowledge (Fig 6).

Univariate analysis using independent-samples t-tests identified several demographic factors associated with knowledge scores. Males demonstrated significantly higher total knowledge scores than females (22.54 ± 7.27 vs. 16.60 ± 6.40;

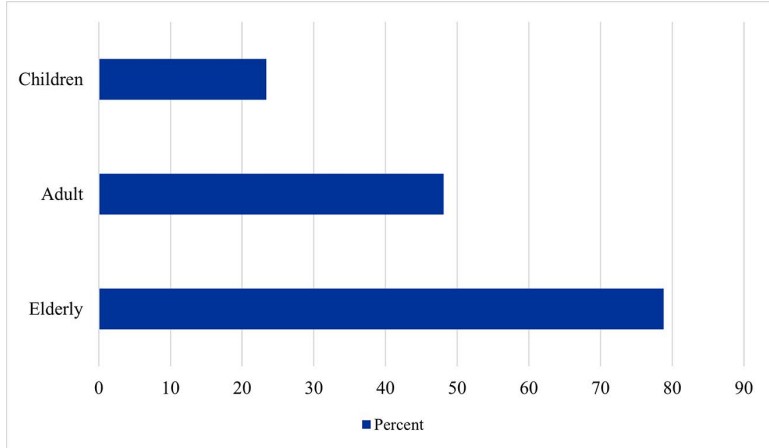

**Fig 1. Most susceptible age group to food–drug interactions according to participants (n = 401).**

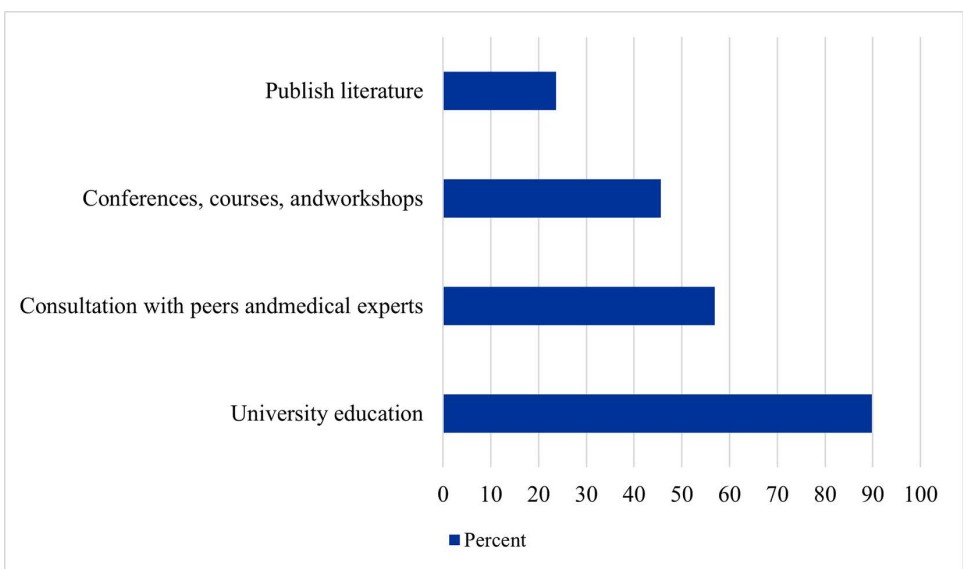

**Fig 2. Main sources of knowledge about food–drug interactions among participants (n = 401).**

mean difference = 5.94; p < 0.001). Pharmacists residing in Abu Dhabi also had higher scores compared with those from other emirates (20.57 ± 7.16 vs. 16.77 ± 7.20; mean difference = 3.90; p = 0.038). Similarly, pharmacists with ≥6 years of experience showed higher knowledge scores than those with ≤5 years of experience (20.67 ± 7.79 vs. 18.57 ± 7.16; mean difference = 2.10; p = 0.001).

No significant differences were observed by age group (19.66 ± 7.15 vs. 18.77 ± 7.57; mean difference = 0.89; p = 0.232) or practice setting (19.63 ± 7.48 vs. 18.28 ± 7.16; mean difference = 1.35; p = 0.164).

According to the multiple linear regression analysis, the total knowledge score was significantly affected by gender, age, years of experience, and whether the participants who self-assessed themselves as well-informed about food-drug interactions (Table 2). Specifically, male pharmacists (p-value <0.001), those aged 20–29 years (p-value = 0.041),

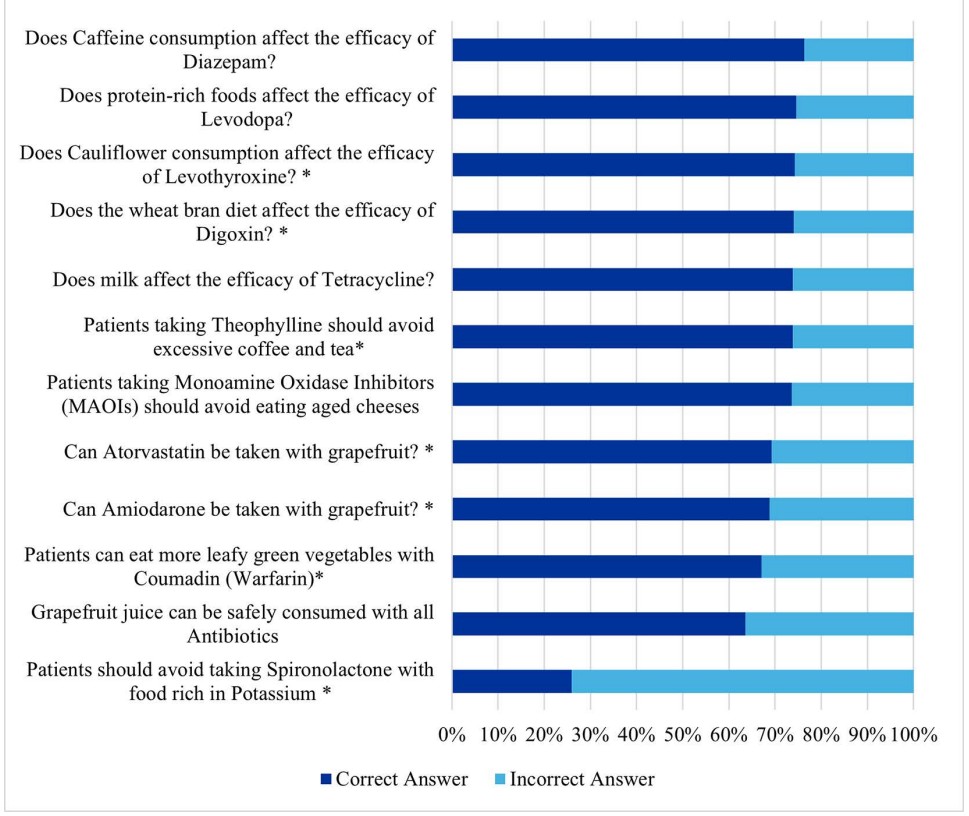

**Fig 3. Participants' responses to 12 items assessing knowledge of drug–food interactions (n = 401).**

pharmacists with ≥6 years of experience (p-value = 0.004), and those who considered themselves well-informed about food-drug interactions (p-value<0.001) demonstrated higher total knowledge scores compared to other participants.

## Discussion

This study evaluated the knowledge and awareness of pharmacists in the United Arab Emirates UAE regarding drug-food and drug-alcohol interactions. To our knowledge, this is the first study addressing pharmacists' knowledge of DFIs/DAIs conducted in the UAE. Although the study design was cross-sectional and descriptive in nature, it provides essential baseline data regarding pharmacists' knowledge in a region where such evidence has been lacking. Establishing this baseline is a necessary first step for designing targeted educational and interventional strategies aimed at improving patient safety. The results of this study indicate that community pharmacists in the UAE have an overall moderate level of knowledge. Scoring 19 out of 30 on average, which aligns with the findings of colleagues in Palestine and Jordan [16,20] suggesting that, despite pharmacists' critical role in maintaining medication safety, there are still gaps in their understanding of clinically significant interactions.

Pharmacists' knowledge was significantly associated with gender, age, years of experience, and self-awareness regarding DFIs. Male pharmacists, younger professionals, and individuals with more than six years of experience showed higher knowledge scores. This is consistent with previous research that demonstrates that demographic and professional factors, including age, employment status, and practical experience affected the healthcare professionals' knowledge of DFIs [19,16,21]. Although age was not significantly associated with knowledge scores in the univariate analysis, it became

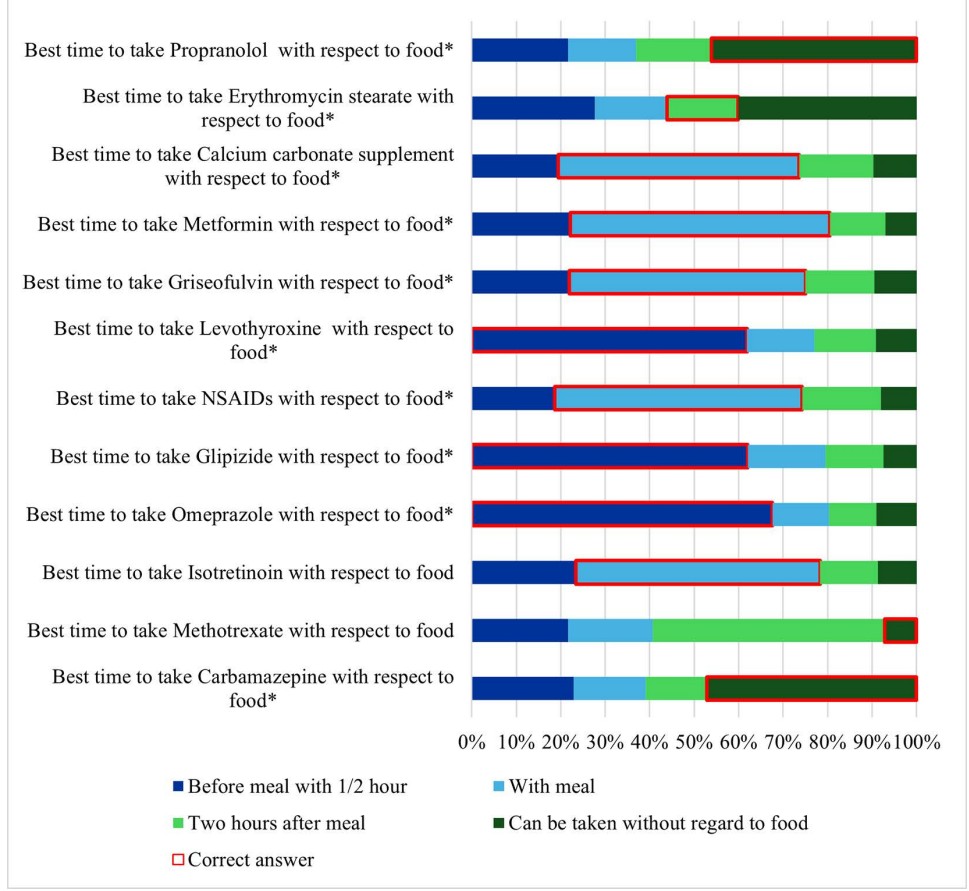

**Fig 4. Participants' responses to 12 items assessing knowledge of drug–food timing (n = 401).**

significant in the multivariable regression model after adjustment for other demographic variables. This suggests that potential confounding factors may influence the relationship between age and knowledge scores. However, given the borderline statistical significance, this finding should be interpreted with caution.

Regarding food–drug interactions, participants demonstrated good awareness of DFIs involving central nervous system (CNS) and narrow therapeutic index drugs, such as Diazepam, MAOI, Digoxin, and Warfarin. The attentiveness of the DFIs to such drugs may be attributed to their early discovery, utilization, and extensive research conducted on them [20,22]. Making them recognizable illustrations of DFIs in both literary contexts and educational institutions, as well as practical environments. [22]. Similarly, pharmacists recognized interactions leading to reduced drug absorption through complex formation, deterring absorption and ultimately resulting in lowering drug bioavailability [20,23]. However, knowledge gaps remained for less frequently recognized interactions, such as those between potassium-rich foods and Spironolactone, highlighting areas for targeted educational improvement.

With respect to drug–alcohol interactions, pharmacists showed greater level of awareness compared to some DFI areas, over three-quarters accurately identified potential interactions between alcohol with methotrexate, warfarin, isoniazid, metformin, paracetamol, and antihistamines. The enhanced awareness may be due to the fact that these medicines are well known for their hepatotoxicity, effects on the CNS, or increased risk of bleeding when used in combination with alcohol. However, given the risk of alcohol consumption poses to public health, it is concerning that some individuals did

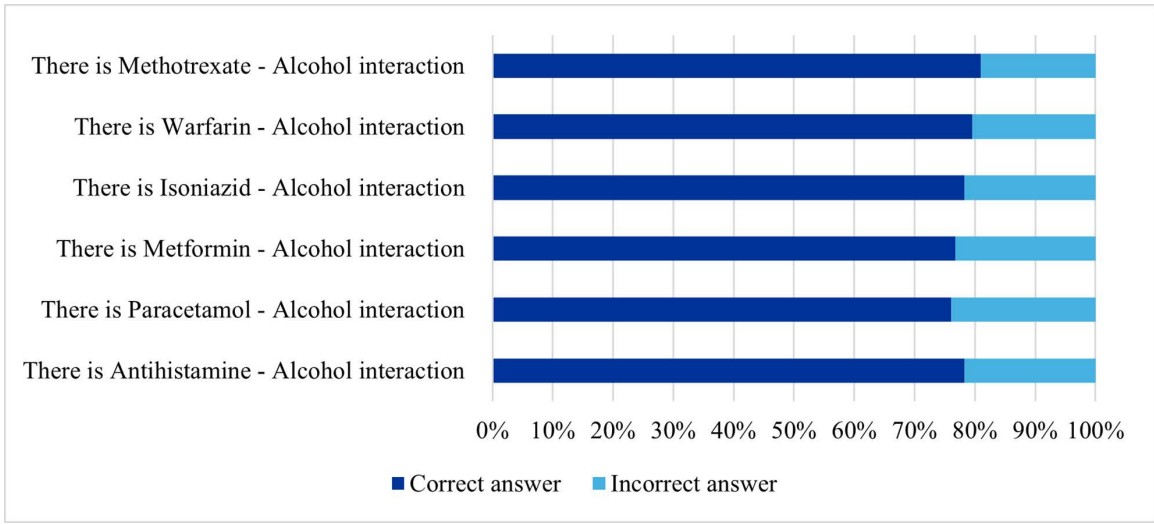

**Fig 5. Participants' responses to 6 items assessing knowledge of drug–alcohol interactions (n=401).**

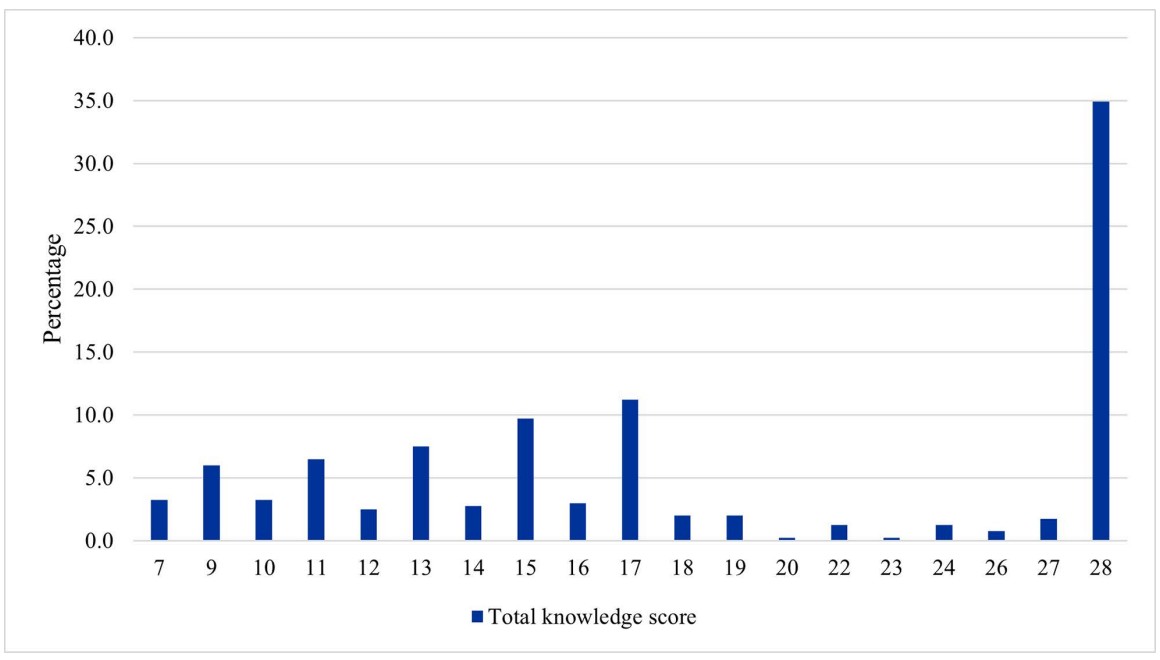

**Fig 6. Distribution of total knowledge scores among participants (n=401).**

not provide accurate answers to these questions indicating that there is potential for improvement. From a clinical perspective, failure to recognize high-risk interactions—such as potassium-rich foods with spironolactone or alcohol with hepatotoxic medications—may expose patients to serious adverse events including hyperkalaemia, bleeding, or liver injury. These gaps therefore represent meaningful patient safety concerns rather than solely academic deficiencies.

**Table 2. Factors associated with total knowledge scores among participants (n = 401).**

| Parameter | Total Score | | | |
|---|---|---|---|---|
| | **Beta** | **P-value#** | **Beta** | **P-value$** |
| **Gender** | | | | |
| • Male | Reference | | | |
| • Female | −0.399 | <0.001 | −0.428 | **<0.001** |
| **Age** | | | | |
| • 20–29 years | Reference | | | |
| • ≥ 30 years | −0.060 | 0.228 | −0.091 | **0.041** |
| **Emirate** | | | | |
| • Abu-Dhabi | Reference | | | |
| • Others | −0.248 | <0.001 | −0.073 | 0.123 |
| **Practice place** | | | | |
| • Chain pharmacy | Reference | | | |
| • Individual pharmacy | −0.087 | 0.083 | 0.053 | 0.246 |
| **Years of experience** | | | | |
| • 0–5 years | Reference | | | |
| • ≥ 6 years | 0.128 | 0.010 | 0.138 | **0.004** |
| **Do you think you have enough information about food-drug interactions?** | | | | |
| • No | Reference | | | |
| • Yes | 0.404 | <0.001 | 0.390 | **<0.001** |

#: Using simple linear regression, $: Using multiple linear regression, *Significant at 0.05 significance level

These results are in line with research conducted both internationally and in other Middle Eastern nations, which indicates that pharmacists and other healthcare professionals have moderate to inadequate awareness of DFIs/DAIs [16,19,20,24,25]. Healthcare workers in these studies usually showed a thorough understanding of high-risk or extensively researched interactions, but they showed limited awareness of less frequent or recently identified interactions. This pattern emphasizes the necessity of ongoing professional development and the inclusion of context-specific, clinically relevant DFI/DAI content in pharmacy education.

## Implications for practice

The present findings carry important implications for pharmacy practice in the UAE. Pharmacists are often the first point of contact for patients and thus play a critical role in counselling about both food–drug and drug–alcohol interactions. Emphasizing the need for structured training and continuous professional development focused on clinically relevant food–drug interactions. Such programs proved to be imperative and effective in improving healthcare providers' knowledge and practices [25]. Therefore, incorporating mandatory DFI/DAI training within pharmacy curricula and structured continuing education programs for practicing pharmacists—across both hospital and community settings—would strengthen professional competencies and promote safer medication use.

## Study limitations

This study has several limitations. First, self-selection bias may have occurred, as pharmacists who were more motivated or interested in DFIs/DAIs may have been more likely to participate. Second, the reliance on self-reported responses introduces the possibility of social desirability bias, although this was eased by ensuring anonymity, which is known to

reduce both social desirability and social anxiety. Third, the study may have been more susceptible to selection and recall biases due to the use of an online survey, which may have influenced the representativeness of the sample. Finally, the results are specific to community pharmacists in the United Arab Emirates and might not apply to hospital pharmacists or other healthcare professionals. Additionally, the knowledge assessment relied on closed-ended, recall-based questions, which may not fully capture applied clinical decision-making or real-world counselling practices. Furthermore, all items were equally weighted despite differences in clinical severity among interactions. Future studies incorporating simulated cases, observational methods, or patient outcomes would provide a more comprehensive evaluation of pharmacists' competencies.

### Future research directions

Future research should examine pharmacists' knowledge and practices in a variety of healthcare contexts, such as clinical settings and hospitals, to capture more comprehensive picture of pharmacists' proficiency in handling DFIs and DAIs. Focus groups and interviews are examples of qualitative research methods that may be used to identify the underlying causes of knowledge gaps and barriers to putting knowledge into practice. Intervention studies are also required to assess how well case-based learning programs, focus active training, and continuing education enhance knowledge and counselling abilities. Lastly, investigating patients' knowledge and awareness of DFIs and DAIs as well as how pharmacists might best assist them will provide important insights into maximizing the safe use of medications in the UAE.

## Conclusion

In conclusion, this study emphasizes the necessity of a comprehensive educational approach. Pharmacists in the UAE can be more capable to recognize, advise on, and prevent harmful food-drug and drug-alcohol interactions by combining formal academic training, continuing professional development, and workplace-based reinforcement. This will ultimately strengthen medication safety and improve healthcare outcomes.

## Supporting information

**S1 Questionnaire. Survey instrument used to assess pharmacists' knowledge of drug–food and drug–alcohol interactions.**
(PDF)

**S1 Data. Anonymized raw dataset underlying the findings of this study.**
(XLSX)

## Author contributions

**Conceptualization:** Taima Qudah, Leen Fino, Razan I. Nassar, Suhad Abumueis.

**Data curation:** Taima Qudah.

**Formal analysis:** Taima Qudah, Razan I. Nassar, Suhad Abumueis.

**Methodology:** Taima Qudah.

**Visualization:** Razan I. Nassar.

**Writing – original draft:** Taima Qudah, Leen Fino, Razan I. Nassar, Suhad Abumueis.

**Writing – review & editing:** Taima Qudah, Leen Fino, Razan I. Nassar, Suhad Abumueis.

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
