## [Decision Letter · Decision Letter 0]

6 Feb 2026

Dear Dr. Abumueis,

Thank you for submitting your manuscript to PLOS ONE. After careful consideration, we feel that it has merit but does not fully meet PLOS ONE’s publication criteria as it currently stands. Therefore, we invite you to submit a revised version of the manuscript that addresses the points raised during the review process.

We look forward to receiving your revised manuscript.

Kind regards,

Mohammed Zawiah

Academic Editor

PLOS One

Journal Requirements:

https://journals.plos.org/plosone/s/file?id=wjVg/PLOSOne_formatting_sample_main_body.pdf and and https://journals.plos.org/plosone/s/file?id=ba62/PLOSOne_formatting_sample_title_authors_affiliations.pdf

3. In the online submission form, you indicated that the data underlying this cross-sectional study are available from the corresponding author upon reasonable request. The dataset contains sensitive participant information, and public sharing is restricted to protect confidentiality. Data will be shared in anonymized form with qualified researchers upon reasonable request.

4. Please remove your figures from within your manuscript file, leaving only the individual TIFF/EPS image files, uploaded separately. These will be automatically included in the reviewers’ PDF**.**

5. Please include your tables as part of your main manuscript and remove the individual files. Please note that supplementary tables (should remain/ be uploaded) as separate "supporting information" files.

Reviewers' comments:

Reviewer's Responses to Questions

**Comments to the Author**

1. Is the manuscript technically sound, and do the data support the conclusions?

Reviewer #1: Yes

Reviewer #2: No

Reviewer #3: Partly

Reviewer #4: Yes

2. Has the statistical analysis been performed appropriately and rigorously?

Reviewer #1: Yes

Reviewer #2: No

Reviewer #3: No

Reviewer #4: Yes

3. Have the authors made all data underlying the findings in their manuscript fully available?

Reviewer #1: Yes

Reviewer #2: Yes

Reviewer #3: Yes

Reviewer #4: Yes

4. Is the manuscript presented in an intelligible fashion and written in standard English?

Reviewer #1: Yes

Reviewer #2: Yes

Reviewer #3: Yes

Reviewer #4: Yes

Reviewer #1: This manuscript addresses an important and clinically relevant topic regarding community pharmacists’ knowledge of drug–food and drug–alcohol interactions in the UAE. The study is well designed, clearly written, and fills a regional research gap, as similar data from the UAE are limited. The s questionnaire shows acceptable reliability, and the statistical analyses are appropriate. Results are presented clearly and discussed in relation to previous regional and international studies.

Some points require clarification. The recruitment through online professional groups may have introduced selection bias, which may have influenced knowledge scores and should be discussed in more depth. The associations between knowledge scores and demographic factors (gender, age, experience) should be interpreted cautiously, with emphasis that causality cannot be inferred. Highlighting the clinical significance of specific knowledge gaps would further strengthen the practical implications.

Ethical approval and consent procedures are clearly reported, and no concerns regarding research or publication ethics were identified.

Overall, this is a valuable contribution and is suitable for publication after minor revisions.

Reviewer #2: The study is purely descriptive and confirmatory, documenting “moderate knowledge” among pharmacists—a conclusion that is already well-established in the literature. There is no hypothesis-driven framework, no conceptual model, and no attempt to link knowledge gaps to measurable patient safety outcomes, dispensing errors, or real-world clinical consequences.

Knowledge was assessed using closed-ended, recall-based questions that test memorization of textbook examples rather than applied clinical decision-making. The scoring system treats all items as equally important, despite major differences in clinical risk (e.g., caffeine–diazepam vs. potassium–spironolactone). No weighting, validation against practice behavior, or triangulation with real counselling practices was performed.

Reviewer #3: The study addresses an important issue but inferential statistics have not been utilized properly. comparison between different genders, graduation year, nationality, age, and location need to be added showing the significance of these factors and reflects the curricular impact of pharmacy practice.

Reviewer #4: The manuscript is well written and developed.

The title is appropriate and matches the objectives of the study.

The methodology is robust.

The findings are valuable and well presented.

The conclusion is well stated, reflecting the results section

A minor revision is required: that is paraphrasing line 10-15 in page 9.

**Do you want your identity to be public for this peer review?** For information about this choice, including consent withdrawal, please see our For information about this choice, including consent withdrawal, please see our Privacy Policy .

Reviewer #1: No

Reviewer #2: No

Reviewer #3: No

Reviewer #4: No

---

## [Author Response · Author response to Decision Letter 1]

10 Feb 2026

Dear Editor, Dear Reviewers,

We would like to thank the reviewers for their valuable comments and suggestions. We addressed all points and revised the script accordingly. Revisions are marked in YELLOW in the script. We downloaded the revised manuscript and used red highlight for the new comments.

We hope that this revision is satisfactory. Thank you.

Reviewer comments

Reviewer #1:

1-The recruitment through online professional groups may have introduced selection bias, which may have influenced knowledge scores and should be discussed in more depth.

2-The associations between knowledge scores and demographic factors (gender, age, experience) should be interpreted cautiously, with emphasis that causality cannot be inferred.

3-Highlighting the clinical significance of specific knowledge gaps would further strengthen the practical implications.

Reviewer #2:

1-The study is purely descriptive and confirmatory, documenting “moderate knowledge” among pharmacists—a conclusion that is already well-established in the literature.

2-There is no hypothesis-driven framework, no conceptual model, and no attempt to link knowledge gaps to measurable patient safety outcomes, dispensing errors, or real-world clinical consequences.

3-Knowledge was assessed using closed-ended, recall-based questions that test memorization of textbook examples rather than applied clinical decision-making.

4-The scoring system treats all items as equally important, despite major differences in clinical risk (e.g., caffeine–diazepam vs. potassium–spironolactone). No weighting, validation against practice behavior, or triangulation with real counselling practices was performed.

Reviewer #3:

The study addresses an important issue but inferential statistics have not been utilized properly. comparison between different genders, graduation year, nationality, age, and location need to be added showing the significance of these factors and reflects the curricular impact of pharmacy practice.

Reviewer #4:

1-A minor revision is required: that is paraphrasing line 10-15 in page 9.

Authors’ response:

Reviewer #1: Thank you very much for your comments.

1-To address the selection bias, we added “This convenience-based online recruitment approach may have preferentially attracted pharmacists who are more professionally engaged or academically inclined, potentially introducing selection bias.” In the method section.

2-knowledge was influenced by gender in in the discussion changed to read as “were significantly associated with” to avoid causal is this correct.

3-paragraph read as “From a clinical perspective, failure to recognize high-risk interactions—such as potassium-rich foods with spironolactone or alcohol with hepatotoxic medications—may expose patients to serious adverse events including hyperkalaemia, bleeding, or liver injury. These gaps therefore represent meaningful patient safety concerns rather than solely academic deficiencies.” Added to the discussion

Reviewer #2: Thank you very much for your comments.

1-We added “Although the study design was cross-sectional and descriptive in nature, it provides essential baseline data regarding pharmacists’ knowledge in a region where such evidence has been lacking. Establishing this baseline is a necessary first step for designing targeted educational and interventional strategies aimed at improving patient safety.” To the first paragraph in the discussion to address the comment.

2-Thank you very much for your comments we added “Guided by the premise that pharmacists’ demographic and professional characteristics may influence their knowledge and counselling practices, we hypothesized that knowledge levels would vary according to age, gender, and years of experience. Identifying these variations may help inform targeted educational strategies and professional development initiatives to enhance medication safety.” In the introduction to address the hypothesis/ framework comment.

3-paragraph read as “The questionnaire primarily assessed factual knowledge using closed-ended items to ensure standardization and objective scoring across respondents. While this approach allows efficient evaluation of baseline knowledge, it may not fully reflect real-world counselling behaviors or clinical decision-making.” added to the method section.

Also, we added “Additionally, the knowledge assessment relied on closed-ended, recall-based questions, which may not fully capture applied clinical decision-making or real-world counselling practices.” To the limitation section

4-Paragraph added to method section read as “All items were weighted equally to maintain consistency and comparability with previous knowledge-based surveys. However, we acknowledge that different interactions may carry varying levels of clinical risk, and future research could consider risk-based weighting or case-based assessment approaches.”

Also, we added “Furthermore, all items were equally weighted despite differences in clinical severity among interactions. Future studies incorporating simulated cases, observational methods, or patient outcomes would provide a more comprehensive evaluation of pharmacists’ competencies.” To the limitation section.

Reviewer #3: Thank you very much for your comments.

Independent-samples t-tests were conducted and paragraph read as

“Independent-samples t-tests were conducted to compare mean knowledge scores across binary demographic and professional variables, including gender, age group, geographical location, pharmacy type, and years of experience. “Added to the method section

Also, the results presented in the result section “Univariate analysis using independent-samples t-tests identified several demographic factors associated with knowledge scores. Males demonstrated significantly higher total knowledge scores than females (22.54 ± 7.27 vs. 16.60 ± 6.40; mean difference = 5.94; p < 0.001). Pharmacists residing in Abu Dhabi also had higher scores compared with those from other emirates (20.57 ± 7.16 vs. 16.77 ± 7.20; mean difference = 3.90; p = 0.038). Similarly, pharmacists with ≥6 years of experience showed higher knowledge scores than those with ≤5 years of experience (20.67 ± 7.79 vs. 18.57 ± 7.16; mean difference = 2.10; p = 0.001).

No significant differences were observed by age group (19.66 ± 7.15 vs. 18.77 ± 7.57; mean difference = 0.89; p = 0.232) or practice setting (19.63 ± 7.48 vs. 18.28 ± 7.16; mean difference = 1.35; p = 0.164).”

Reviewer #4: Thank you very much for your comments.

The specified section has been paraphrased for clarity to read as “highlighting the imperative role of training pharmacists and the incorporation of educational curricula and continuous educational programs for clinically relevant food-drug interactions.”

Journal requirements

1. PLOS formatting style

All manuscript formatting has been revised to comply with PLOS ONE style guidelines.

2–3. Data sharing policy and Data Availability statement

As there are no ethical or legal restrictions preventing data sharing, we prepared and uploaded an anonymized raw dataset.

• Uploaded dataset as Supporting Information (S1 Data.xlsx)

• Uploaded the full questionnaire as Supporting Information (S1 Questionnaire.pdf)

• Updated the Data Availability statement in the manuscript to:

All relevant data underlying the findings of this study are provided within the manuscript and its Supporting Information files.

4. Figures

All figures have been:

• Removed from the main manuscript

• Saved individually in TIFF format

• Uploaded separately as Fig1–Fig6 according to PLOS naming requirements

5. Tables

All tables:

• Included directly within the manuscript

• Presented as editable Word tables

• Positioned immediately after first citation

• Not uploaded as separate files

---

## [Decision Letter · Decision Letter 1]

10 Mar 2026

Dear Dr. Abumueis,

Thank you for submitting your manuscript to PLOS ONE. After careful consideration, we feel that it has merit but does not fully meet PLOS ONE’s publication criteria as it currently stands. Therefore, we invite you to submit a revised version of the manuscript that addresses the points raised during the review process.

**ACADEMIC EDITOR:**

Thank you for the revision. The manuscript has improved. Please address the new comment from Reviewer 2. In addition, two important points still need revision from my side:

Please clarify the age-related findings. In univariate analysis, age was not significant (p = 0.232), but in multiple regression age 20–29 years became significant (p = 0.041). That can happen after adjustment, but the manuscript does not explain it. Since this is a borderline result, it needs cautious interpretation.Please strengthen the limitation on sampling bias. The manuscript acknowledges possible selection bias from online recruitment, but this is later minimized. Please keep the limitation wording balanced and transparent.

We look forward to receiving your revised manuscript.

Kind regards,

Mohammed Zawiah

Academic Editor

PLOS One

Journal Requirements:

Reviewers' comments:

Reviewer's Responses to Questions

**Comments to the Author**

Reviewer #2: All comments have been addressed

Reviewer #3: All comments have been addressed

2. Is the manuscript technically sound, and do the data support the conclusions?

Reviewer #2: Partly

Reviewer #3: Yes

3. Has the statistical analysis been performed appropriately and rigorously?

Reviewer #2: I Don't Know

Reviewer #3: Yes

4. Have the authors made all data underlying the findings in their manuscript fully available?

Reviewer #2: Yes

Reviewer #3: Yes

5. Is the manuscript presented in an intelligible fashion and written in standard English?

Reviewer #2: No

Reviewer #3: Yes

Reviewer #2: - Alcohol use prevalence is required to be added in the introduction.

- All comment have been addressed.

- English proofreading is required.

Reviewer #3: I recommend therticle to be Accepted as it is since The author added all of the required corrections.

**Do you want your identity to be public for this peer review?** For information about this choice, including consent withdrawal, please see our For information about this choice, including consent withdrawal, please see our Privacy Policy .

Reviewer #2: No

Reviewer #3: **Yes:** Muaed Jamal AlomarMuaed Jamal Alomar

---

## [Author Response · Author response to Decision Letter 2]

12 Mar 2026

Dear Academic Editor and Reviewers,

We would like to thank the Academic Editor and reviewers for their valuable comments and suggestions. We addressed all points and revised the script accordingly. All changes are highlighted in the tracked-changes version of the manuscript. Our point-by-point responses are provided below.

We hope that this revision is satisfactory. Thank you.

Academic Editor Comments

1. Please clarify the age-related findings. In univariate analysis, age was not significant (p = 0.232), but in multiple regression age 20–29 years became significant (p = 0.041). That can happen after adjustment, but the manuscript does not explain it. Since this is a borderline result, it needs cautious interpretation.

2. Please strengthen the limitation on sampling bias. The manuscript acknowledges possible selection bias from online recruitment, but this is later minimized. Please keep the limitation wording balanced and transparent.

Reviewer #2:

1. Alcohol use prevalence is required to be added in the introduction.

2. English proofreading is required.

Authors’ response:

Academic Editor: Thank you very much for your comments.

1. We have clarified this point in the Discussion section and added an explanation that the association became significant after adjustment for other variables in the multivariable regression model. We added new sentences to the discussion read as “Although age was not significantly associated with knowledge scores in the univariate analysis, it became significant in the multivariable regression model after adjustment for other demographic variables. This suggests that potential confounding factors may influence the relationship between age and knowledge scores. However, given the borderline statistical significance, this finding should be interpreted with caution.”

2. We have revised the limitations section to present the potential sampling bias more transparently. The sentence: “however, as pharmacists are increasingly using smartphones and the internet in their work, this is unlikely to have had an impact on our findings.” Deleted and replaced with: “Third, the study may have been more susceptible to selection and recall biases due to the use of an online survey, which may have influenced the representativeness of the sample.”

Reviewer #2: Thank you very much for your comments.

1-“Although alcohol consumption in the United Arab Emirates is regulated, evidence indicates that alcohol use remains present and represents a public health concern in the country [7], underscoring the importance of awareness of potential drug–alcohol interactions in clinical practice.” Added to the introduction to address the comment.

2- Proofreading is done. Examples are below:

Introduction

• Current

For example, Interactions between alcohol and antihistamines, as well as nonsteroidal anti-inflammatory medicines (NSAIDs), elevate the risk of internal bleeding, and respiratory difficulties. or cause life-threatening cardiovascular complications when taken with monoamine oxidase inhibitors (MAOIs).

Replaced with:

For example, interactions between alcohol and antihistamines or nonsteroidal anti-inflammatory drugs (NSAIDs) may increase the risk of internal bleeding and respiratory complications. Alcohol may also cause life-threatening cardiovascular complications when combined with monoamine oxidase inhibitors (MAOIs).

• Current

medication review [13]According to international guidelines

Space added

medication review [13]. According to international guidelines

• Current

While excessive intake of vitamin K–rich foods diminish warfarin's safety and effectiveness

Replaced with

Excessive intake of vitamin K–rich foods diminishes warfarin's safety and effectiveness

Results

• Current

"Does protein-rich foods affect the efficacy of Levodopa?"

Replaced with

"Do protein-rich foods affect the efficacy of levodopa?"

Discussion

• Current

The results of this study revealed that pharmacists in the UAE have overall moderate knowledge among UAE pharmacists.

Replaced with:

The results of this study indicate that community pharmacists in the UAE have an overall moderate level of knowledge.

• Current

This is consistent with previous research that demonstrate

Replaced with:

This is consistent with previous research that demonstrates

• Current

UAE regarding drug interactions

To read better:

UAE regarding drug–food and drug–alcohol interactions

Limitations

• Current

second, the reliance on self-reported responses

Replaced with:

Second, the reliance on self-reported responses

Future research

• Current

lastly, investigating patients' knowledge

Replaced with:

Lastly, investigating patients' knowledge

---

## [Editor Report · Decision Letter 2]

16 Mar 2026

Drug–food and drug–alcohol interactions: pharmacists’ knowledge gaps and patient safety concerns in the United Arab Emirates

PONE-D-25-61208R2

Dear Dr. Abumueis,

We’re pleased to inform you that your manuscript has been judged scientifically suitable for publication and will be formally accepted for publication once it meets all outstanding technical requirements.

Kind regards,

Mohammed Zawiah

Academic Editor

PLOS One
---

## [Editor Report · Acceptance letter]

PONE-D-25-61208R2

PLOS One

Dear Dr. Abumueis,

I'm pleased to inform you that your manuscript has been deemed suitable for publication in PLOS One. Congratulations! Your manuscript is now being handed over to our production team.

Kind regards,

on behalf of

Dr. Mohammed Zawiah

Academic Editor

PLOS One